# Mental health and wellbeing outcomes of youth participation: A scoping review protocol

**Marlee Bower**[1]*, **Amarina Donohoe-Bales**[1], **Andre Quan Ho Nguyen**[1], **Scarlett Smout**[1], **Julia Boyle**[1], **Emma Barrett**[1], **Stephanie R. Partridge**[2,3], **Mariam Mandoh**[2], **Magenta Simmons**[4,5], **Radhika Valanju**[6], **Fulin Yan**[6], **Cheryl Ou**[6], **Danica Meas**[6], **Kailin Guo**[6], **Dominik Mautner**[6], **Imeelya Al Hadaya**[6], **Dominique Rose**[6], **Maree Teesson**[1]

**1** Matilda Centre for Research in Mental Health and Substance Use, Faculty of Medicine and Health, The University of Sydney, Camperdown, New South Wales, Australia, **2** Engagement and Codesign Hub, School of Health Sciences, Faculty of Medicine and Health, The University of Sydney, Camperdown, New South Wales, Australia, **3** Charles Perkins Centre, The University of Sydney, Camperdown, New South Wales, Australia, **4** Orygen, The National Centre of Excellence in Youth Mental Health, Parkville, Victoria, Australia, **5** Centre for Youth Mental Health, The University of Melbourne, Parkville, Victoria, Australia, **6** Youth Mental Health Advisory Team (YMHAT), Matilda Centre for Research in Mental Health and Substance Use, Faculty of Medicine and Health, The University of Sydney, Camperdown, New South Wales, Australia

* marlee.bower@sydney.edu.au

**Data Availability Statement:** No datasets were generated or analysed during the current study. All relevant data from this study will be made available upon study completion.

## Abstract

There is growing recognition that young people should be given opportunities to participate in the decisions that affect their lives, such as advisory groups, representative councils, advocacy or activism. Positive youth development theory and sociopolitical development theory propose pathways through which youth participation can influence mental health and wellbeing outcomes. However, there is limited empirical research synthesising the impact of participation on youth mental health and/or wellbeing, or the characteristics of activities that are associated with better or worse mental health and/or wellbeing outcomes. This scoping review seeks to address this gap by investigating the scope and nature of evidence detailing how youth participation initiatives can influence mental health and/or wellbeing outcomes for participants. To be eligible, literature must describe youth (aged 15–24) in participation activities and the impact of this engagement on participant mental health and/or wellbeing outcomes. A systematic scoping review of peer-reviewed and grey literature will be conducted using Scopus, PsycINFO, Embase, Medline and grey literature databases. The scoping review will apply established methodology by Arksey and O'Malley, Levac and colleagues and the Joanna Briggs Institute. Title, abstract, and full text screening will be completed by two reviewers, data will be extracted by one reviewer. Findings will be reported in accordance with the Preferred Reporting Items for Systematic reviews and Meta-Analyses extension for Scoping Reviews (PRISMA-ScR), including a qualitative summary of the characteristics of youth participation and their influence on youth mental health outcomes. Youth advisory group members will be invited to deliver governance on the project from the outset; participate in, and contribute to, all stages of the review process; reflect on their own experiences of participation; and co-author the resulting publication. This scoping review will provide essential knowledge on how participation activities can be better designed to maximise beneficial psychosocial outcomes for involved youth.

**Funding:** This research is supported via seed funding from the National Health and Medical Research Council Centre of Research Excellence in Prevention and Early Intervention in Mental Illness and Substance Use (PREMISE: APP11349009, awarded to MB) and the BHP Foundation (IRMA#: G209081, awarded to MT). The funders had and will not have a role in study design, data collection and analysis, decision to publish, or preparation of the manuscript.

**Competing interests:** MB, ADB, SS and MT are supported by the BHP Foundation. This does not alter our adherence to PLOS ONE policies on sharing data and materials. To our knowledge, remaining authors report no known conflicts of interest for this paper.

## Introduction

The period of 'youth', defined here as 15 to 24 years-of-age, [1] is characterised by significant social and developmental change, as young people transition away from the limited and dependent roles of childhood and adolescence towards the formation of distinct identities in emerging adulthood [2]. During this period, young people experience rapid cognitive and physiological changes while developing critical social skills, knowledge and networks that enable them to engage with broader society [3]. Approximately 75% of diagnosed mental disorders initiate before the age of 25 and have long-lasting social, health, and economic impacts for the individual, their families, and society [4]. As such, this represents a critical period for engagement in activities that promote mental health and wellbeing.

Since the United Nations Convention on the Rights of the Child, [5] there has been growing emphasis on addressing the challenge of young people's participation rights. This challenge is particularly relevant to young people's right to have a voice and engage in 'meaningful participation', whereby young people can express their views, have their voices heard, and are involved in key decision-making processes in matters relevant to them [6–8]. Youth-led organisations have highlighted that when young people feel unheard in relation to local and global concerns pertinent to their lives, it can cause stress and compound anxiety associated with these issues [9]. Therefore, participation may produce psychological benefits for young people [10, 11].

Youth engagement and/or participation is defined as the process of engaging young people in the institutions, issues and decisions that affect their lives [6]. Participation may take on a variety of forms including, but not limited to, advocacy, advisory, activism, decision-making or civic engagement activities in research or policy-making settings. Participation can differ according to young peoples' level of engagement, agency and influence in decision-making, including passive consultative participation to more active, collaborative, youth-led and co-produced participation [12, 13]. Participation does not include non-genuine forms of engagement, whereby people in power impose agendas on and/or aim to educate or treat young participants [14]. Several theories propose how participation may contribute to improved mental health and wellbeing outcomes. These include Positive Youth Development (PYD) theory and Sociopolitical Development (SPD) theory [15].

PYD theory posits that positive experiences in developmental contexts (such as communities, schools, and families) enable healthy youth development. Participation in extracurricular activities in these settings is associated with empowerment through greater self-esteem and social support, and fewer depressive symptoms and risk-taking behaviours [16]. As such, youth engagement in advisory, advocacy or decision-making capacities may also improve mental health through empowerment [16]. SPD theory focuses on the context of oppression and disadvantage regarding the evolving critical understanding of political, cultural, economic, and other systemic forces that shape society and the individual within it. Within this framework, activism and resistance may serve as a particularly important medium for young people to engage with socio-political systems and promote healthy development. Ballard [15] also notes that youth civic engagement can promote skills and attributes associated with resilience, indicating that meaningful youth participation may serve as a protective factor against adverse mental health impacts of challenging life experiences.

There is growing evidence that responding to traumatic events by engaging in positive activities like advocacy efforts, volunteering or altruism can foster post-traumatic growth and positively impact mental wellbeing. In a study of Australian community members, involvement in voluntary groups–comprising civic participation and engagement activities–led to a reduction in individual and community-level post-traumatic stress symptoms in the years

following a major bushfire disaster [17]. More recently, participation in mutual aid community support groups during the COVID-19 pandemic was associated with positive emotional wellbeing, improved relationships, and greater sense of control [18]. There is also some evidence that opportunities for active engagement and climate action may mitigate young peoples' distress associated with climate change [9]. These findings show promise for the role of participation and engagement activities by positively impacting youth participant mental health and wellbeing, particularly in post-disaster and climate contexts. Although beyond the scope of the present review, it is worth noting that there is a substantial body of literature on the benefits for broader groups who are the recipients and beneficiaries of programs and initiatives that have had meaningful youth participation [19, 20].

This scoping review is not limited to positive mental health and wellbeing impacts of youth participation. Some components of youth participation could be harmful for participant mental health and wellbeing. These include youth participation activities/programs that are overly 'extractive' rather than collaborative in nature; overburdening or setting unrealistic expectations of participants; [21] treating involvement as tokenistic; [22] or not taking participants' input seriously [23, 24]. Additionally, activities that require participants to share information about difficult or traumatic life experiences [23] or exposure to sensitive materials without adequate support or debriefing may also compromise emotional wellbeing [25].

When considering the potential mental health effects of youth participation, it is important to consider and differentiate between various forms of engagement. For instance, volunteer activities which allow young people to focus their energy into individual actions to alleviate the suffering of others may influence health through the positive feelings associated with helping others. In contrast, political activism in the form of protesting, building coalitions, or petitioning may influence health through empowerment [15].

There is currently a gap for synthesis of the existing body of literature examining the mental health and/or wellbeing outcomes of youth participation. In particular, there is limited synthesis around the causal or directional evidence between youth participation and mental health, as well as pathways and mechanisms through which youth engagement may influence mental health outcomes. A comprehensive synthesis of the literature is needed to understand the characteristics and social determinants of young people who benefit from, or who are harmed by, different forms of youth engagement, including those from marginalised groups such as culturally and linguistically diverse communities; lesbian, gay, bisexual, transgender, queer, intersex, asexual and other sexually or gender diverse (LGBTQIA+) peoples; and those with low socioeconomic status. This evidence would provide essential knowledge on how these activities could be better designed to maximise beneficial outcomes for involved youth.

A preliminary search of existing registries and databases were conducted and no published or underway systematic reviews or scoping reviews on this topic were identified. As such, this scoping review seeks to address this gap in research by investigating the scope and nature of evidence detailing how youth participation initiatives can influence mental health outcomes and/or wellbeing outcomes for participants.

## Materials and methods

A scoping review was chosen, in place of a systematic review, to preliminarily assess the size and scope of available peer-reviewed and grey literature pertaining to youth participation and mental health and/or wellbeing. This scoping review will conform to the reporting standards of the Preferred Reporting Items for Systematic reviews and Meta-Analyses extension for Scoping Reviews (PRISMA-ScR) and the Joanna Briggs Institute guidelines for scoping reviews [26]. This scoping review will be conducted in accordance with Levac's

recommendations [27] and Arksey and O'Malley's [28] six-stage methodology for scoping reviews: 1) identifying research questions; 2) searching relevant literature; 3) selecting evidence; 4) charting the data; 5) collating and reporting the results; and 6) consulting with stakeholders to inform study findings. This is a flexible and iterative approach to systematically scoping and interpreting available literature.

This protocol has been registered within the Open Science Framework (available at https://osf.io/2qfy7) and has been reported in accordance with the Preferred Reporting Items for Systematic Reviews and Meta-Analyses Protocols (PRISMA-P) statement (S1 Table).

Due to the subject area of the scoping review, it was deemed important to include young people throughout the review process. An advisory group of eight 15–24 year-olds with experience of mental health concerns and/or participation, named the Youth Mental Health Advisory Team (YMHAT), was established. As part of the selection process for the YMHAT, an expression of interest form and project description were emailed to members of existing youth advisory groups connected to the research team's institutions and those interested came on board. The YMHAT were initially consulted to develop and refine the search strategy and will be consulted henceforth to gain their feedback and expertise at each stage of the project, build on results, co-author publications, and disseminate findings. Interested YMHAT members were included as part of the protocol authorship team. As an acknowledgment of the valuable time and expertise generously contributed by YMHAT members to the project, and their ongoing support, compensation for their time spent was provided in the form of a gift voucher valued at $30 per hour. This compensation amount was determined based on local government guideline rates for youth lived experience expertise, in alignment with best practices. Ethical approval was not required, as YMHAT members were engaged in an advisory and co-author capacity, not as study participants.

Stage 1: Identifying the research questions

Research questions were drafted by the research team according to the population, intervention and outcome (PIO) format. For further information about the PIO format and study context, see Stage 3: Evidence selection. Research questions were then workshopped and refined iteratively with the YMHAT. During this consultation, it was agreed that youth participation can encompass advisory, advocacy or decision-making activities.

1. What is the evidence for associations between youth participation and mental health and/or wellbeing outcomes for participants?

2. What are the components or processes of youth participation activities that promote or diminish mental health and/or wellbeing?

3. What are the evidence gaps in the literature examining youth mental health and/or wellbeing outcomes related to participation activities?

Stage 2: Searching relevant literature

As recommended by the Joanna Briggs Institute (JBI), [26] a three-stage search strategy will be used. First, a basic preliminary search was conducted through the Scopus and Medline databases which returned multiple relevant studies. Key search terms, described below, were refined through analysis of the keywords in the title and abstracts of relevant articles, and of the index terms used to describe these articles. The search strategy and selected keywords, including a combination of free text terms and Medical Subject Headings (MeSH) terms, were then refined based on consultation with the YMHAT and two experienced health and social science librarians at the University of Sydney. Because youth participation is multidisciplinary by nature and often occurs outside of academic spheres (e.g., policy-making), the search

strategy will be purposively broad, including multidisciplinary databases and both academic and grey literature.

Key search terms include:

i. mental health terms: mental health/disorders, depression, anxiety, distress, suicide, empowerment, hope, confidence, happiness, wellbeing, self-efficacy;

ii. participation terms: leadership, decision-making, participation; civic/political engagement, activism, advisory, policy-making, advocacy, co-production/design;

iii. youth terms: young adult, young people, youth, teen, adolescence; child; and

iv. experience terms: feedback, experience, benefit, perspective.

Secondly, all identified keywords and index terms will be systematically searched across four major peer-reviewed databases: Medline (through PubMed), Scopus, PsycInfo, and Embase. See S2 Table for an example of a preliminary database search using the finalised search terms.

A modified keyword-only strategy will be used for grey literature: defined as research that is produced outside of traditional academic publishing channels, including government reports and policy papers. Grey literature will be sourced from Google Scholar, Google, CADTH, EThOS, HMIC, OpenGrey, WHO and Analysis and Policy Observatory (APO) databases. The search results in Google Scholar will be limited to the first 4 pages of the search, with 50 results per page. Grey literature keywords include: civic engagement, empowerment, mental health, young people, participation, collaborative engagement, co-production/design, protest, political participation, youth advisory board/structure, representative council, advisory, participatory design, youth voice, citizenship, youth rights and consumer involvement.

Thirdly, the reference lists of identified literature will be searched for additional sources.

Stage 3: Evidence selection

Inclusion criteria

Inclusion criteria

*Participants*. The United Nations (UN) defines youth as the period of transition from the dependence of childhood towards the independence of adulthood [1]. While recognising that the term 'youth' is fluid and varies in different societies around the world, the UN defines youth as persons between the ages of 15 and 24 for statistical purposes. For a comprehensive analysis of the existing body of research, this scoping review will assess literature in which youth are defined as aged 15 to 24.

*Concept/Intervention*. The primary intervention is youth participation or engagement, including but not limited to decision-making, advisory or advocacy activities, both formal and informal in nature. Participation and engagement were defined as activities that aim to impact youth as a collective, rather than on an individual level, thus excluding shared decision-making literature [29]. For example, studies detailing engaging in a school representative council or going to a protest to advocate for broader climate change action would be included in this study. In contrast, studies describing engaging in shared-decision making with a healthcare provider in personal treatment decisions would be excluded because the impact is individual rather than collective.

*Outcomes*. To be eligible for inclusion, studies must explore the association between youth participation and mental health and/or wellbeing outcomes for participants. Mental health outcomes include psychological distress, general stress, anxiety disorders, mood disorders, post-traumatic stress, post-traumatic growth, trauma and sleep disorders. Wellbeing outcomes include general wellbeing, resilience, self-efficacy, self-esteem, empowerment, quality of life, confidence and hope.

*Context.* As the aim of the scoping review is to understand how youth participation influences mental health and wellbeing outcomes for participants, the review will include literature from a broad context. Literature published in English examining all countries, sexual orientation and gender identities, marginalised groups, and culturally and linguistically diverse groups, excluding those without an abstract, will be considered.

*Study designs.* In recognition of the trans-disciplinary nature of youth participation work, this review will consider a broad range of study designs including, but not limited to, quantitative, qualitative and mixed-methods studies, including experimental, quasi-experimental (i.e. studies that aim to evaluate interventions but do not use randomisation), observational, longitudinal and cross-sectional designs. Grey literature outputs will include government publications, policy statements, theses, conference proceedings, non-peer reviewed research reports and white papers. Commentary and opinion papers will be excluded in this scoping review as they do not meet the criteria for evidence concerning the effect of youth participation on mental health and/or wellbeing outcomes.

Following the search, results will be collated and uploaded into EndNote X9 and duplicates removed. Using Covidence software, all titles, abstracts and full-texts will be assessed against the inclusion criteria by two reviewers. Potentially relevant literature will be retrieved, and their citation details uploaded onto the JBI system for Unified Management [30]. Reasons for the exclusion of texts at this stage will be recorded and reported in the scoping review. Disagreement between reviewers at any stage of the screening process will be resolved through discussion.

Stage 4: Charting the data

Information from eligible literature will be charted using the JBI data extraction tool. The information extracted will include specific details about the authors, year of publication, study title and objectives, participants, type of participation, mental health and/or wellbeing outcomes assessed, context, study methods and key findings relevant to the review question/s. The data extraction tool will be pretested by two reviewers against a small sample of included literature to ensure that the tool is adequately capturing all relevant data. The remaining data will be extracted by one reviewer. Any modifications made to the extraction tool during the review process will be detailed in the resultant scoping review. If appropriate, authors of papers will be contacted to request missing or additional data, where required.

It is anticipated that data will be extracted and interpreted by exploring how different forms and components of youth participation and engagement relate to different mental health and/or wellbeing outcomes for participants. For example, exploring how impacts on mental health may differ according to different types of participation, including formal (e.g., an advisory group) versus informal participation (e.g., attending a protest) or when adult-led versus youth-led; or 'components' of specific participation programs, such as using in-person or virtual participation techniques or the types of post-program feedback mechanisms employed. Results may be extracted and interpreted according to the characteristics of youth participation each study explores and/or according to the type of mental health outcome reported (e.g., anxiety or empowerment).

Stage 5: Collating and reporting the results

Search results and screening steps will be detailed in a PRISMA flowchart and reporting of findings will be informed by the PRISMA-ScR Checklist. Findings from the scoping review will be presented in graphical and tabular format. Narrative synthesis will be adopted to further elaborate on qualitative findings and describe how results relate to, and answer, the research objectives. Quantitative data will be analysed by descriptive numerical summary and qualitative content analysis techniques. Evidence synthesis, interpretation and conclusions made will

be informed by the UNICEF Guidelines on Adolescent Participation and Civic Engagement [12] and the positive youth development 5Cs framework [31].

Stage 6: Consulting with stakeholders

Consultations will be conducted iteratively with the YMHAT throughout stages of the scoping review. The YMHAT has assisted in refining research questions and search terms based on their feedback and expertise. The YMHAT will remain engaged during screening, data extraction and data analysis stages, and they will help to reflect on research findings and identify gaps and priorities for future research in the mental health-related youth participation space. YMHAT members will be supported and guided by the wider research team and provided with sufficient training to undertake research activities. At present, members of the YMHAT have reported positive experiences from their involvement in the project as advisors and co-authors, including increased confidence and motivation, feeling heard, and greater sense of purpose.

## Discussion

This scoping review will provide valuable evidence on the link between participation and mental health and/or wellbeing outcomes for involved youth. In particular, this review aims to understand which components of participation, and under which conditions, contribute to poorer or better mental health for youth participants. As far as the author team are aware, this will be the first scoping review to comprehensively synthesise a broad range of peer-reviewed and grey literature at the intersection between youth participation and mental health and wellbeing. Findings from this review will form a strong foundation for future research and an important evidence base from which participation activities can be better designed and evaluated to ensure beneficial psychosocial outcomes for young people. Furthermore, the purposeful inclusion of the YMHAT–a diverse group of young people with differing lived experiences of mental health and participation–will provide greater depth and rich perspectives to the research project, while strengthening the relevance and accessibility of the review findings. Members of the YMHAT will mutually benefit from skills gained in the research process, increased knowledge and greater sense of empowerment [32]. However, the scoping review has some limitations. Language limits may prevent inclusion of potentially relevant non-English studies and, despite an inclusive and comprehensive search strategy, some studies may be missed due to the interdisciplinary nature of the review. Lastly, the incorporation of the YMHAT requires considerable preparation, training and renumeration costs which may delay the scoping review process. Nevertheless, these processes are critical in achieving safe, inclusive, fair and positive youth participation outcomes, while increasing the likelihood of achieving more impactful research.

The findings of this scoping review will be disseminated through peer-reviewed publications and conferences, as well as relevant stakeholder groups. The YMHAT will be consulted to identify additional dissemination avenues to enhance research accessibility, particularly for young readers.

Any changes to the review protocol will be reported in the final scoping review.

## Supporting information

**S1 Table. PRISMA-P (Preferred Reporting Items for Systematic review and Meta-Analysis Protocols) checklist.**
(DOCX)

**S2 Table. Preliminary database search strategy.**
(DOCX)

## Acknowledgments

The authors would like to thank Kanchana Ekanayake and Bernadette Carr from the University of Sydney Library for their support with developing the search strategy and the YMHAT for their contribution to the design and development of the broader research project.

## Author Contributions

**Conceptualization:** Marlee Bower, Amarina Donohoe-Bales, Andre Quan Ho Nguyen, Scarlett Smout, Julia Boyle, Emma Barrett, Stephanie R. Partridge, Mariam Mandoh, Magenta Simmons, Radhika Valanju, Fulin Yan, Cheryl Ou, Danica Meas, Kailin Guo, Dominik Mautner, Imeelya Al Hadaya, Dominique Rose, Maree Teesson.

**Data curation:** Marlee Bower, Amarina Donohoe-Bales, Andre Quan Ho Nguyen, Scarlett Smout, Julia Boyle, Emma Barrett, Stephanie R. Partridge, Mariam Mandoh, Magenta Simmons, Radhika Valanju, Fulin Yan, Cheryl Ou, Danica Meas, Kailin Guo, Dominik Mautner, Imeelya Al Hadaya, Dominique Rose, Maree Teesson.

**Funding acquisition:** Marlee Bower.

**Investigation:** Marlee Bower, Amarina Donohoe-Bales, Andre Quan Ho Nguyen, Scarlett Smout, Julia Boyle, Emma Barrett, Stephanie R. Partridge, Mariam Mandoh, Magenta Simmons, Radhika Valanju, Fulin Yan, Cheryl Ou, Danica Meas, Kailin Guo, Dominik Mautner, Imeelya Al Hadaya, Dominique Rose, Maree Teesson.

**Methodology:** Marlee Bower, Amarina Donohoe-Bales, Andre Quan Ho Nguyen, Scarlett Smout, Julia Boyle, Emma Barrett, Stephanie R. Partridge, Mariam Mandoh, Magenta Simmons, Radhika Valanju, Fulin Yan, Cheryl Ou, Danica Meas, Kailin Guo, Dominik Mautner, Imeelya Al Hadaya, Dominique Rose, Maree Teesson.

**Project administration:** Marlee Bower, Amarina Donohoe-Bales, Andre Quan Ho Nguyen, Scarlett Smout, Maree Teesson.

**Resources:** Marlee Bower, Amarina Donohoe-Bales, Andre Quan Ho Nguyen, Scarlett Smout, Julia Boyle, Emma Barrett, Stephanie R. Partridge, Mariam Mandoh, Magenta Simmons, Radhika Valanju, Fulin Yan, Cheryl Ou, Danica Meas, Kailin Guo, Dominik Mautner, Imeelya Al Hadaya, Dominique Rose, Maree Teesson.

**Supervision:** Marlee Bower.

**Visualization:** Marlee Bower, Amarina Donohoe-Bales, Andre Quan Ho Nguyen, Scarlett Smout, Julia Boyle, Emma Barrett, Stephanie R. Partridge, Mariam Mandoh, Magenta Simmons, Radhika Valanju, Fulin Yan, Cheryl Ou, Danica Meas, Kailin Guo, Dominik Mautner, Imeelya Al Hadaya, Dominique Rose, Maree Teesson.

**Writing – original draft:** Marlee Bower, Amarina Donohoe-Bales, Andre Quan Ho Nguyen, Scarlett Smout.

**Writing – review & editing:** Marlee Bower, Amarina Donohoe-Bales, Andre Quan Ho Nguyen, Scarlett Smout, Julia Boyle, Emma Barrett, Stephanie R. Partridge, Mariam Mandoh, Magenta Simmons, Radhika Valanju, Fulin Yan, Cheryl Ou, Danica Meas, Kailin Guo, Dominik Mautner, Imeelya Al Hadaya, Dominique Rose, Maree Teesson.

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
