## [Decision Letter · Decision Letter 0]

12 Apr 2023

PONE-D-22-34628

Mental health and wellbeing outcomes of youth participation: A scoping review protocol

PLOS ONE

Dear Dr. Donohoe-Bales,

Thank you for submitting your manuscript to PLOS ONE. After careful consideration, we feel that it has merit but does not fully meet PLOS ONE’s publication criteria as it currently stands. Therefore, we invite you to submit a revised version of the manuscript that addresses the points raised during the review process.

I. To revise according to the reviewers suggestion attached in the manuscript. 

We look forward to receiving your revised manuscript.

Kind regards,

Kofi Aduo-Adjei

Academic Editor

PLOS ONE

Journal Requirements:

“MB, ADB, SS and MT are supported by the BHP Foundation. To our knowledge, remaining authors report no known conflicts of interest for this paper.”

Reviewers' comments:

Reviewer's Responses to Questions

**Comments to the Author**

1. Does the manuscript provide a valid rationale for the proposed study, with clearly identified and justified research questions?

Reviewer #1: Yes

Reviewer #2: Yes

2. Is the protocol technically sound and planned in a manner that will lead to a meaningful outcome and allow testing the stated hypotheses?

Reviewer #1: Yes

Reviewer #2: Yes

3. Is the methodology feasible and described in sufficient detail to allow the work to be replicable?

Reviewer #1: Yes

Reviewer #2: Yes

4. Have the authors described where all data underlying the findings will be made available when the study is complete?

Reviewer #1: Yes

Reviewer #2: Yes

5. Is the manuscript presented in an intelligible fashion and written in standard English?

Reviewer #1: Yes

Reviewer #2: Yes

6. Review Comments to the Author

You may also provide optional suggestions and comments to authors that they might find helpful in planning their study.

Reviewer #1: Overall this is a well drafted scoping review protocol, and the proposed study is of importance to the youth globally. I have picked a few minor issues for your attention; Line 32, LGBTQIA+ should be written in full for the first time. The scoping review question should capture the population(p), concept (c) and context (c), it seems the context is not stated in your question. Consider presenting the final main search string from your preliminary search - lines 176 & 177. I have noted the involvement of the youth in this study, present ethical clearance reference for this study and explain the selection process for the advisory group. What are the strengths, limitations and de-limitations for the proposed study? Make sure your reference list has document identification numbers (DOI)

Reviewer #2: This is an excellent protocol! I really love that the team is including youth throughout the scoping review process. I have no comments.

7. PLOS authors have the option to publish the peer review history of their article (what does this mean?). If published, this will include your full peer review and any attached files.

Reviewer #1: No

Reviewer #2: No

<quillbot-extension-portal></quillbot-extension-portal>

---

## [Author Response · Author response to Decision Letter 0]

6 Jun 2023

The following responses to reviewer and editor comments are included in the attached 'Response to Reviewers' document. 

Journal Requirements:

Author response: The author team have updated the manuscript to reflect PLOS ONE’s style requirements, as per the ‘Title, author, affiliations formatting guidelines’ and the ‘Manuscript body formatting guidelines’. 

2. Please confirm that the competing interest statement does not alter your adherence to all PLOS ONE policies on sharing data and materials, by including the following statement: "This does not alter our adherence to PLOS ONE policies on sharing data and materials.” Please include your updated Competing Interests statement in your cover letter; we will change the online submission form on your behalf.

Author response: The authors confirm that the competing interest statement does not alter adherence to PLOS ONE policies. Please update our competing interest statement to the following: 

“MB, ADB, SS and MT are supported by the BHP Foundation. This does not alter our adherence to PLOS ONE policies on sharing data and materials. To our knowledge, remaining authors report no known conflicts of interest for this paper.”

Author response: The author team have updated the manuscript to reflect PLOS ONE’s reference style. The authors have downloaded and used the PLOS reference style for EndNote.

Reviewer’s Comments

Reviewer #1: Overall this is a well drafted scoping review protocol, and the proposed study is of importance to the youth globally. I have picked a few minor issues for your attention; 

1) Line 132, LGBTQIA+ should be written in full for the first time. 

2) The scoping review question should capture the population(p), concept (c) and context (c), it seems the context is not stated in your question. 

3) Consider presenting the final main search string from your preliminary search - lines 176 & 177. 

4) I have noted the involvement of the youth in this study, present ethical clearance reference for this study and explain the selection process for the advisory group. 

5) What are the strengths, limitations and de-limitations for the proposed study? 

6) Make sure your reference list has document identification numbers (DOI)

Author response to Reviewer 1: 

1) The acronym ‘LGBTQIA+’ is now written in full in lines 133-134. 

2) The research questions were developed in accordance with the participant, intervention and outcome (PIO) format, hence the context (C) is not stated, nor is it relevant to our scoping review questions. We have clarified this in lines 172-174, flagging that further information about the study ‘context’ is provided in Stage 3: Evidence selection. 

3) We thank the reviewer for this suggestion and have made sure to clearly indicate where this is included in the supplementary materials (see lines 202-203). Due to the substantial word count in the final search string, we are unable to include it in the body of the manuscript. However, we have provided a condensed version of the final search terms in lines 194-200. 

4) Young people were involved in this project in an advisory and co-author capacity (as part of the Youth Mental Health Advisory Team), not as study participants, and hence no ethical clearance or approval is required for their involvement. We have clarified this, as well as details regarding the selection process for the advisory group, remuneration costs and their key responsibilities, in lines 157-170 and lines 292-294. 

5) Strengths and limitations of the proposed scoping review have now been added to the discussion section in lines 300-316.

6) Digital object identifiers (DOIs) have now been added to the reference list. 

Reviewer #2: This is an excellent protocol! I really love that the team is including youth throughout the scoping review process. I have no comments.

Author response to Reviewer 2: N/A

---

## [Decision Letter · Decision Letter 1]

4 Oct 2023

Mental health and wellbeing outcomes of youth participation: A scoping review protocol

PONE-D-22-34628R1

Dear Dr. Marlee Bower,

We’re pleased to inform you that your manuscript has been judged scientifically suitable for publication and will be formally accepted for publication once it meets all outstanding technical requirements.

Kind regards,

Joseph Adu, PhD

Academic Editor

PLOS ONE

Additional Editor Comments (optional):

Reviewers' comments:

Reviewer's Responses to Questions

**Comments to the Author**

1. Does the manuscript provide a valid rationale for the proposed study, with clearly identified and justified research questions?

Reviewer #2: Yes

2. Is the protocol technically sound and planned in a manner that will lead to a meaningful outcome and allow testing the stated hypotheses?

Reviewer #2: Yes

3. Is the methodology feasible and described in sufficient detail to allow the work to be replicable?

Reviewer #2: Yes

4. Have the authors described where all data underlying the findings will be made available when the study is complete?

Reviewer #2: Yes

5. Is the manuscript presented in an intelligible fashion and written in standard English?

Reviewer #2: Yes

6. Review Comments to the Author

You may also provide optional suggestions and comments to authors that they might find helpful in planning their study.

Reviewer #2: I have no comments. The authors have addressed reviewer feedback very well.

7. PLOS authors have the option to publish the peer review history of their article (what does this mean?). If published, this will include your full peer review and any attached files.

Reviewer #2: No

---

## [Editor Report · Acceptance letter]

9 Oct 2023

PONE-D-22-34628R1 

Mental health and wellbeing outcomes of youth participation:
A scoping review protocol 

Dear Dr. Donohoe-Bales:

I'm pleased to inform you that your manuscript has been deemed suitable for publication in PLOS ONE. Congratulations! Your manuscript is now with our production department. 

Kind regards, 

on behalf of

Dr Joseph Adu 

Academic Editor

PLOS ONE